# The Use of Halophytic Companion Plant (*Portulaca oleracea* L.) on Some Growth, Fruit, and Biochemical Parameters of Strawberry Plants under Salt Stress

**Sema Karakas** [1,*]**, Ibrahim Bolat** [2] **and Murat Dikilitas** [3]

1. Department of Soil Science and Plant Nutrition, Faculty of Agriculture, Harran University, Sanliurfa 63300, Turkey
2. Department of Horticulture, Faculty of Agriculture, Harran University, Sanliurfa 63300, Turkey; ibolat@harran.edu.tr
3. Department of Plant Protection, Faculty of Agriculture, Harran University, Sanliurfa 63300, Turkey; m.dikilitas@harran.edu.tr
* Correspondence: skarakas@harran.edu.tr

**Abstract:** Strawberry is a salt-sensitive plant adversely affected by slightly or moderately saline conditions. The growth, fruit, and biochemical parameters of strawberry plants grown under NaCl (0, 30, 60, and 90 mmol L$^{-1}$) conditions with or without a halophytic companion plant (*Portulaca oleracea* L.) were elucidated in a pot experiment. Salt stress negatively affected the growth, physiological (stomatal conductance and electrolyte leakage), and biochemical parameters such as chlorophyll contents (chl-*a* and chl-*b*); proline, hydrogen peroxide, malondialdehyde, catalase, and peroxidase enzyme activities; total soluble solids; and lycopene and vitamin C contents, as well as the mineral uptake, of strawberry plants. The companionship of *P. oleracea* increased fresh weight, dry weight, and fruit average weight, as well as the total fruit yield of strawberry plants along with improvements of physiological and biochemical parameters. This study showed that the cultivation of *P. oleracea* with strawberry plants under salt stress conditions effectively increased strawberry fruit yield and quality. Therefore, we suggest that approaches towards the use of *P. oleracea* could be an environmentally friendly method that should be commonly practiced where salinity is of great concern.

**Keywords:** abiotic stress; strawberry; companion plants; phytoremediation

## 1. Introduction

Salinity is one of the most devastating environmental problems limiting crop productivity and quality in many regions of the world. This problem is more prevalent in arid and semi-arid climatic regions. It affects approximately 20% of the cultivated and 50% of irrigated agricultural lands [1,2]. It has now been estimated that 1.5 million hectares of lands have been lost every year due to salinity problems. If salinization goes with this trend, nearly 50% of cultivable lands will be lost by the mid-point of this century [3,4].

Salinity negatively affects plant growth in terms of osmotic, ionic, and nutrient imbalance [5]. These disorders cause oxidative stress on plants. If plants cannot get enough water under high salt stress, turgor pressure significantly decreases, and thus the closure of the stomata of plants becomes inevitable to conserve water [6]. This significantly affects the photosynthetic capacity of plants. Ionic toxicity, on the other hand, inhibits cellular metabolism and biochemical pathways. For example, Na$^+$ ions at the root cell disturb enzymatic activities and inhibit the uptake of other minerals such as K$^+$ and Ca$^{++}$ [7]. The high accumulation of Na$^+$ and Cl$^-$ ions result in many morphological, physiological, molecular, and biochemical pathways in plants. Due to disturbed mechanisms in plants, NaCl stress leads to the development of leaf chlorosis and necrosis, as well as the loss of quality in crops. As a consequence, the assimilation of carbohydrates and sugars allocated

for fruit development is significantly reduced due to stress development and defense mechanisms [8,9].

Plants can be divided into halophytes and glycophytes as responses to salinity stress. Most glycophytes are salt-sensitive, even at low concentrations, while halophytes are highly salt-tolerant plants, which enables them to survive and thrive in extremely saline environments [10,11]. Salt ions have to be taken up by halophytes and deposited in the vacuoles of leaf or root tissues or in separate organelles. In general, salt secretion takes place through the shedding of salty leaves and salt glands or specialized leaf cells [12]. Most halophytes are able to survive by maintaining negative water potential under extreme salt concentrations. Therefore, a true halophyte is considered to maintain its viability and complete the life cycle at NaCl levels between 200 and 1000 mmol $L^{-1}$. These concentrations are very close to the concentrations of seawater level. Some halophytes, on the other hand, tolerate much higher concentrations of NaCl [10,13]. Halophytes such as *Atriplex* spp., *Chenopodium* spp., *Portulaca* spp., *Suaeda* spp., and *Salsola* spp. uptake salt ions through their roots and store them in their leaves. It is quite possible that these plants could be used as companion plants with crop plants, especially salt-sensitive glycophytes, to reduce the negative effects of NaCl through the uptake of toxic ions [14,15]. For example, *Portulaca oleracea* L. (purslane) (which is a member of Portulacaceae), is a drought- and salt-tolerant annual plant. The plant is a promising crop species in saline–alkali soils [16,17]. Moreover, *P. oleracea* could effectively absorb salts from soil media to remediate saline–alkali soils [18]. Previous studies have investigated the effects of salinity on *P. oleracea* growth. For example, Grieve and Suarez [19] evaluated *P. oleracea* responses with saline irrigation, and they showed that the plant could survive at a salinity of 28.5 dS $m^{-1}$. The authors further elucidated the performances of salt-tolerant halophyte species of *P. oleracea* and *Salsola soda* against increasing NaCl levels. They reported that *P. oleracea* and *S. soda* seeds were effectively germinated between 250 and 350 mmol $L^{-1}$ NaCl levels [20].

Strawberry is an economically important fruit crop that is globally cultivated. It belongs to the *Fragaria* genus in Rosaceae family, which contains 23 species [21,22]. The popularity of strawberry fruit crops is increasing in the world due to increasing consumption. Its popularity is also increasing along with the generation of new varieties. Strawberry cultivation has therefore become an important greenhouse and open field crop in the Mediterranean area, although drought and salinity have played significant roles in limiting crop production [23,24].

Strawberry is considered to be sensitive to NaCl salinity due to increased osmotic pressure and $Na^+$ or $Cl^-$ ion toxicity. NaCl salinity not only reduces the crop yield but also deteriorates the quality parameters in many crops, including strawberries [25,26].

In the present study, we elucidated the effects of different NaCl concentrations on strawberry plants grown with or without the halophytic companion plant *P. oleracea* L. to remediate the physiological and biochemical parameters of strawberry plants.

## 2. Materials and Methods

### 2.1. Experimental Design and the Growth of Plants

The experiment was performed between September 2018 and January 2019 in a semi-controlled greenhouse at the University of Harran, Sanliurfa, Turkey. Fresh strawberry (Rubygem variety) plants were grown alone or in combination with *P. oleracea* seedlings in 8-L pots containing peat (Klasmann TS 1) under natural light conditions. Peat is a very porous substrate with an excellent water capacity. Its slow degradation rate, high porosity, and high-water holding capacity makes it one of the most commonly used growth medium, especially for saline-related studies in vegetables and ornamental plants [27,28]. It has low nutrient values, so it is highly unlikely to affect the mineral uptake of macro and micronutrient elements.

The average temperatures for day and night were 35 ± 2/28 ± 2 °C during the course of the experiment. The trial was carried out in a randomized block design. The first group of strawberry plants grown under differing NaCl conditions (0, 30, 60, and

90 mmol L$^{-1}$) was designated as S$_0$, S$_{30}$, S$_{60}$, and S$_{90}$; the second group of plants grown with *P. oleracea* under the same NaCl conditions was designated as SP$_0$, SP$_{30}$, SP$_{60}$, and SP$_{90}$. Treatments in each group were replicated five times. Seedlings were individually transplanted to the pots. Strawberry seedlings following one week of establishment growth in pots were accompanied with *P. oleracea* seeds that were sown and germinated at a rate of 25 companion plants per pot. After germinations (five weeks), the pots were irrigated with or without salt to the full pot capacity throughout the treatment period (twelve weeks). The plants were fertigated with Hoagland's nutrient solution once a week. The experimental trial from the very beginning of obtaining strawberry seedlings to the end of harvest took five months. The plants were harvested at the optimum stage of physiological maturity for the evaluation of salinity responses (Figure 1).

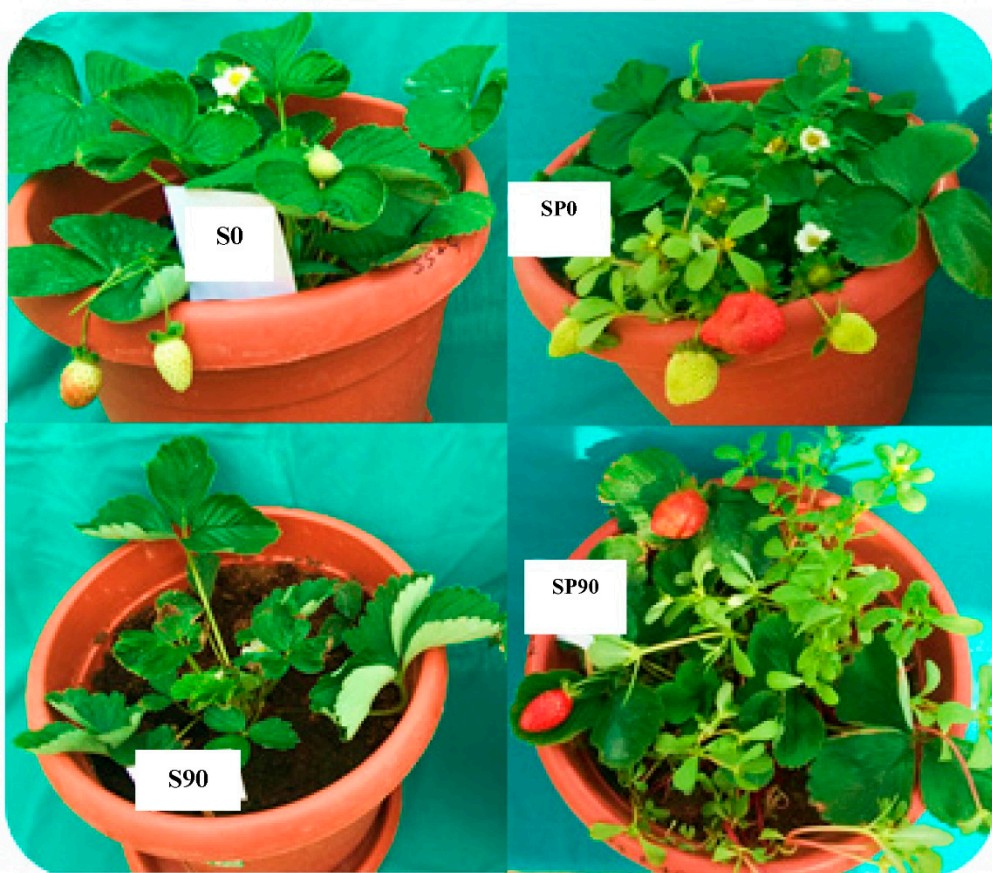

**Figure 1.** Strawberry plants were grown with or without *Portulaca oleracea* under different NaCl conditions.

### 2.2. Plant Growth and Fruit Properties

Strawberry fruits were harvested when 90% of the fruit surface had reached a fully red color. At the end of the experimental period, total fruit weights were determined and the average fruit yield was calculated.

Plant crown and root fresh weight (Fwt) were analyzed immediately after the harvest. The dry weight (Dwt) of plant organs was determined following the drying of plant samples at 70 °C until a constant weight.

Total soluble solids (TSS) were assessed from the fruit juice with a hand refractometer. The results are expressed in percent (%) Catania et al. [29].

The lycopene content of strawberry fruits was assessed according to the method of Barrett and Anthon [30] with minor modifications [31]. One gram of strawberry fruit was homogenized with 10 mL of an ethanol:hexane solution (4:3). The mixture was then

centrifuged at $10,000 \times g$ for 10 min at room temperature. The supernatant (100 µL) was added to 7 mL of the ethanol:hexane solution (4:3) mixture and vortexed. After 1 h of incubation at room temperature, 1 mL of $H_2O$ was added to the tubes and vortexed. The tubes were incubated in the dark to form two different phases. The top phase was taken and read at 503 nm against a hexane blank with a UV microplate spectrophotometer (Epoch, SN: 1611187, Winooski, VT, USA). The lycopene content was calculated according to the following formula (Equation (1)).

$$\mu\text{g Lycopene g}^{-1}\text{ Fwt } = \frac{A_{503} \times 2.7}{172 \times 0.1} \times 537 \qquad (1)$$

where 537 g/mole is the molecular weight of lycopene, 2.7 mL is the volume of the hexane layer, 172 mmol$^{-1}$ is the extinction coefficient for lycopene in hexane, and 0.1 g is the weight of the strawberry.

The vitamin C content of strawberry fruits was assessed according to the method of Oz [32] with small modifications [31]. Strawberry fruits (5 g) were extracted with 25 mL of oxalic acid. The mixture was centrifuged at $10,000 \times g$ for 10 min. Then, 1 mL of this mixture was added to 7 mL of a 1% oxalic acid solution and 8 mL of a dye reagent. The dye reagent was prepared according to the recipe of [31]. The mixture was filtered through Whatman No.2 filter paper and diluted to 100 mL with deionized $H_2O$. Then, 25 mL of this solution were taken and diluted to 500 mL with deionized $H_2O$, vortexed, and kept at 4 °C until use. The mixture was once more vortexed before measurement at 518 nm against the oxalic acid and dye mixture with a UV microplate spectrophotometer (Epoch, SN: 1611187, Winooski, VT, USA).

Electrolyte leakage (EL) was assessed following the method of Lutts et al. [33] using leaf discs for each treatment. Fully expanded young leaves were cleaned three times with deionized $H_2O$ to remove dust and surface-adhered electrolytes. Leaf discs were placed in closed vials containing 10 mL of $H_2O$ and incubated at 25 °C on a rotary shaker for 24 h; subsequently, the electrical conductivity of the solution ($EC_1$) was measured. The final electrical conductivity ($EC_2$) was determined following the autoclaving of the leaf samples at 120 °C for 20 min. Leaf samples were then equilibrated at 25 °C, and the EL was calculated as follows (Equation (2)).

$$EL\ (\%) = \frac{EC1}{EC2} \times 100 \qquad (2)$$

Stomatal conductivity (SC) was determined on the youngest fully expanded leaves on the upper branches of the strawberry plants with a leaf promoter (SC−1) at midday. Measurements were performed by clamping the leaves in the leaf chamber. The actual vapor flux from the leaf through the stomata is expressed as mmol m$^{-2}$ s$^{-1}$, following the work of Karlidag et al. [34].

### 2.3. Biochemical Parameters

Strawberry plant leaf chlorophyll content (Chl-*a* and Chl-*b*) was extracted following the method of Arnon [35] with minor modifications [31]. A sample of the fresh leaf (0.5 g) was homogenized with 10 mL of an acetone:water (80/20, *v/v*) mixture and filtered through Whatman No.2 filter paper and then put into the dark tubes. The Chl-*a* and Chl-*b* contents of the filtrate was measured with a UV microplate spectrophotometer (Epoch, SN: 1611187, Winooski, VT, USA) at 663 and 645 nm, respectively, against an 80% acetone blank. The findings were expressed as mg L$^{-1}$ and calculated as mg g$^{-1}$ Fwt.

The proline concentration was determined following the method of Bates et al. [36] with minor modifications [31]. Leaf samples (0.5 g) were extracted with 10 mL of 3% *w/v* sulphosalicylic acid using a mortar and a pestle. The extract was filtered through Whatman No.2 filter paper. Then, the 2 mL filtrate was mixed with 2 mL of acid–ninhydrin in a test tube and boiled at 100 °C for 1 h. The reaction was terminated in an ice bath. Then, the mixture was extracted using 5 mL of toluene. The tubes were vortexed for 20 s and then

left for 20 min at room temperature to achieve two layers of separation. The organic phase was collected, and the absorbance of the extracts was read at 515 nm using a toluene blank. Proline concentration was made from a standard curve using L-proline (Sigma-Aldrich, Taufkirchen-Germany). The results are expressed as $\mu$mol g$^{-1}$ Fwt.

Hydrogen peroxide ($H_2O_2$) levels were assessed following the method of Velikova et al. [37] with small modifications [38]. Leaf samples (0.5 g) were extracted with 5 mL of 0.1% (*w:v*) trichloroacetic acid (TCA). The extract was centrifuged at 12,000$\times$ *g* at 4 °C for 15 min, and the supernatant (0.5 mL) was added to 0.5 mL of a 10 mmol L$^{-1}$ potassium phosphate buffer (pH 7.0) and 1 mL of a 1 mol L$^{-1}$ potassium iodide. The absorbance was read at 390 nm in a UV microplate spectrophotometer (Epoch, SN: 1611187, Winooski, VT, USA). The $H_2O_2$ content was expressed as $\mu$mol g$^{-1}$ Fwt.

The malondialdehyde (MDA) content was assessed following the method of Sairam and Sexena [39] with minor modifications [38]. The leaf samples (0.5 g) were extracted with 10 mL of a 0.1% (*w/v*) TCA solution. The extract was centrifuged at 10,000$\times$ *g* for 5 min. Four milliliters of 20% *v/v* TCA containing 0.5% *v/v* thiobarbituric acid (TBA) was added to 1 mL of the supernatant. The mixture was kept in boiling water for 30 min, and then the reaction was stopped in an ice bath. The mixture was once more centrifuged at 10,000$\times$ *g* for 5 min and then read in a UV microplate spectrophotometer (Epoch, SN: 1611187, Winooski, VT, USA) at 532 and 600 nm. The MDA content was calculated and expressed as nmol g$^{-1}$ Fwt (Equation (3)).

$$\text{MDA} \left( \text{nmol g}^{-1} \right) = \frac{\text{Extract volume (ml)} \times \left[ (A_{532} - A_{600}) / \left( 155 \text{ mM}^{-1} \text{ cm}^{-1} \right) \right]}{\text{Sample amaunt (g)}} \times 10^3 \tag{3}$$

Catalase enzyme activity (CAT; Enzyme Code. 1.11.1.6) was determined following the method of Milosevic and Slusarenko [40] with minor modifications [38]. Leaf samples (0.5 g) were extracted with 10 mL of a 50 mmol L$^{-1}$ Na-phosphate buffer solution, and then 50 mL of the extract were added to a 2.95 mL reaction mixture (50 mmol L$^{-1}$ Na-phosphate buffer, 10 mmol L$^{-1}$ $H_2O_2$, and 4 mmol L$^{-1}$ Na$_2$EDTA) and read with a UV microplate spectrophotometer (Epoch, SN: 1611187, Winooski, VT, USA) at 240 nm for 30 s. One CAT unit (U) was defined as an increase in absorbance of 0.1 at 240 nm. The activity is expressed as enzyme unite mg$^{-1}$ Fwt.

Peroxidase enzyme activity (POX; Enzyme Code. 1.11.1.7) was assayed following the method of Cvikrova et al. [41] with minor modifications [38]. For the analysis, 100 mL of the homogenate (obtained as above) was added to 3 mL of the reaction mixture (50 mmol L$^{-1}$ Na-phosphate, 5 mmol L$^{-1}$ $H_2O_2$, 13 mmol L$^{-1}$ guaiacols, and pH 6.5). Activity was defined by the range of change in absorbance at 470 nm with a UV microplate spectrophotometer (Epoch, SN: 1611187, Winooski, VT, USA). One unit of POX was defined as a change of 0.1 absorbance unit per minute at 470 nm. Activity is expressed as enzyme unit mg$^{-1}$ Fwt.

Leaf mineral (K$^+$, Na$^+$, Ca$^{2+}$, Mg$^{2+}$, and Cl$^-$) contents were determined according to the procedure made by Chapman and Pratt [42] with minor modifications [31]. Dry plant samples (0.5 g) were ground in porcelain crucibles. The porcelain crucibles were placed into a muffle furnace, and the temperature was gradually increased up to 500 °C. Following cooling, the ash was dissolved in 5 mL of 2 N hydrochloric acid. After 30 min, the volume was made up to 50 mL with distilled $H_2O$, and the supernatant was filtered through Whatman No.42 filter paper. The resulting supernatant containing Na$^+$, K$^+$, Ca$^{+2}$, and Mg$^{+2}$ ions were assessed by Inductively Coupled Plasma (ICP, Perkin Elmer). Chloride was determined using ion chromatography after the filtration through Whatman No.42 filter paper.

Duncan's multiple range test (DMRT) was used to evaluate the data using SPSS 22 (ANOVA test) at a significance level of $p \leq 0.05$ using. Data are presented as a mean value $\pm$ with standard error.

## 3. Results

Strawberry plant growth, fruit properties, biochemical parameters, and mineral contents were significantly affected by all salinity levels. The crown fresh and dry weights of strawberry plants in saline conditions were significantly lower in plants grown alone in saline conditions when compared to those of plants grown in combination with *P. oleracea* under the same conditions. For example, the crown fresh weights of the plants were 55.16, 37.62, and 35.16 g plant$^{-1}$ grown alone in $S_{30}$, $S_{60}$, and $S_{90}$ mmol L$^{-1}$ NaCl conditions, respectively. When plants were grown in combinations with *P. oleracea*, their conditions were significantly improved at all NaCl conditions. The fresh weights of plants increased to 64.38, 44.76, and 44.49 plant$^{-1}$ at the $SP_{30}$, $SP_{60}$, and $SP_{90}$ mmol L$^{-1}$ NaCl conditions, respectively (Table 1). Similar improvements were recorded for the dry weights of plants (Table 1). In general, the combination of companion plants (*P. oleracea*) was found to be effective in increasing the Fwt and Dwt under each NaCl condition.

**Table 1.** Growth and physiological parameters of strawberry plants grown alone or in combination with *P. oleracea* at differing NaCl levels.

| Treatments | Crown Fwt (g Plant$^{-1}$) | Crown Dwt (g Plant$^{-1}$) | EL (%) | SC (mmol m$^{-2}$ s$^{-1}$) |
|---|---|---|---|---|
| $S_0$ | 85.32 ± 4.17 [a] | 18.39 ± 0.70 [a] | 11.90 ± 0.77 [e] | 241.98 ± 4.29 [a] |
| $S_{30}$ | 55.16 ± 5.30 [b] | 12.71 ± 1.40 [b] | 15.61 ± 0.57 [c] | 183.26 ± 8.19 [c] |
| $S_{60}$ | 37.62 ± 2.70 [d] | 10.00 ± 1.07 [d] | 21.84 ± 1.06 [b] | 127.64 ± 8.39 [d] |
| $S_{90}$ | 35.16 ± 1.90 [d] | 9.39 ± 0.69 [d] | 25.34 ± 0.92 [a] | 94.10 ± 3.83 [e] |
| $SP_0$ | 85.57 ± 4.46 [a] | 19.09 ± 0.91 [a] | 11.07 ± 0.45 [e] | 260.38 ± 8.81 [a] |
| $SP_{30}$ | 64.38 ± 5.01 [b] | 14.77 ± 1.08 [b] | 11.10 ± 0.95 [e] | 230.80 ± 5.48 [b] |
| $SP_{60}$ | 44.76 ± 1.67 [c] | 11.23 ± 0.22 [c] | 13.84 ± 1.04 [d] | 181.60 ± 4.65 [c] |
| $SP_{90}$ | 44.49 ± 2.69 [c] | 11.16 ± 0.59 [c] | 15.32 ± 1.03 [c] | 131.08 ± 5.51 [c] |

Significance level at $p \leq 0.05$ was determined for the salt treatment using Duncan's multiple range test. Different letters in each column indicate statistical differences. S: a strawberry grown alone; SP: the strawberry and *P. oleracea* companionship; EL: electrolyte leakage; SC: stomatal conductivity.

EL is considered an important criterion for salt stress parameters. EL was increased with increasing levels of salt. For example, leaf EL was found to be 11.90 and 11.07% at $S_0$ and $SP_0$, respectively. Increases of EL were trended from 15.61 to 25.34% with respect to conditions from $S_{30}$ to $S_{90}$, respectively. When *P. oleracea* was accompanied with strawberry plants in the NaCl conditions, the increase of EC was so minimal that only 11.10 and 15.32% were recorded at $SP_{30}$ and $SP_{90}$, respectively; see Table 1.

Stomatal conductivity in saline conditions was gradually decreased as the concentration of NaCl increased in plants grown alone in saline conditions (Table 1). However, the cultivation of *P. oleracea* improved the SC of strawberry plants under all NaCl conditions when compared to those grown alone in saline conditions. The improvement of SC was evident in that the increases were from 183.26 to 230.80% from $S_{30}$ to $SP_{30}$ cultivation conditions, respectively. At the higher NaCl concentrations of $S_{90}$ and $SP_{90}$, the SC was still improved with a lesser efficiency from 94.10 to 131.08%, respectively.

The average fruit weight and yield of strawberry plants under NaCl conditions were reduced in plants grown alone, but the co-cultivation of strawberry plants with *P. oleracea* increased the average and total fruit weight (Table 2).

The employment of *P. oleracea* not only increased the crop yield and physiological parameters but also improved the quality of fruits in terms of lycopene and vitamin C contents. Lycopene and vitamin C contents were gradually decreased as the concentration of NaCl increased. Again, the employment of *P. oleracea* increased the lycopene and vitamin C at all NaCl levels (Table 2). For example, the remarkable effect was more evident at the 90 mmol L$^{-1}$ NaCl conditions, as the both lycopene and vitamin C contents were increased when grown with *P. oleracea* as compared to those of plants grown alone in saline conditions.

**Table 2.** Yield and some fruit properties of strawberry plants grown alone or in combination with *P. oleracea* at differing NaCl levels.

| Treatments | Average Fruit Weight (g Plant$^{-1}$) | Yield (g Plant$^{-1}$) | Lycopene (mg kg$^{-1}$ Fwt) | Vitamin C (mg kg$^{-1}$ Fwt) | TSS (%) |
|---|---|---|---|---|---|
| $S_0$ | 18.53 ± 0.24 [a] | 214.76 ± 25.26 [a] | 37.98 ± 1.61 [a] | 49.87 ± 2.48 [a] | 8.80 ± 0.22 [a] |
| $S_{30}$ | 14.96 ± 0.58 [c] | 132.85 ± 19.44 [c] | 35.27 ± 1.29 [a] | 45.07 ± 2.55 [c] | 6.80 ± 0.25 [b] |
| $S_{60}$ | 9.80 ± 1.25 [e] | 83.76 ± 15.20 [d] | 27.52 ± 1.13 [b] | 36.79 ± 1.30 [d] | 5.80 ± 0.24 [d] |
| $S_{90}$ | 5.60 ± 0.58 [f] | 31.53 ± 5.40 [e] | 16.56 ± 1.61 [c] | 32.53 ± 0.81 [e] | 5.20 ± 0.25 [e] |
| $SP_0$ | 19.09 ± 0.52 [a] | 229.40 ± 17.46 [a] | 37.20 ± 1.45 [a] | 51.88 ± 1.89 [a] | 9.00 ± 0.20 [a] |
| $SP_{30}$ | 16.94 ± 0.54 [b] | 164.80 ± 23.99 [b] | 34.87 ± 1.95 [a] | 47.19 ± 1.04 [a] | 7.40 ± 0.17 [b] |
| $SP_{60}$ | 15.74 ± 0.53 [c] | 147.57 ± 25.67 [c] | 33.93 ± 1.82 [a] | 41.36 ± 1.70 [c] | 5.90 ± 0.20 [c] |
| $SP_{90}$ | 12.65 ± 0.73 [d] | 94.00 ± 8.91 [d] | 28.55 ± 1.31 [b] | 43.12 ± 2.51 [c] | 5.60 ± 0.19 [c] |

Significance level at $p \leq 0.05$ was determined for the salt treatment using Duncan's multiple range test. Different letters in each column indicate statistical differences. S: a strawberry grown alone; SP: the strawberry and *P. oleracea* companionship; TSS: total soluble solids.

Unlike other parameters, the TSS contents of the fruits in saline conditions were significantly lowered. The co-cultivation of *P. oleracea* did not significantly improve the conditions of strawberry plants (Table 2).

Chl-*a* and Chl-*b* were significantly affected by salinity at the $S_{60}$ and $S_{90}$ mmol L$^{-1}$ NaCl levels ($p \leq 0.05$). For example, the Chl-*a* and Chl-*b* were determined as 0.70 and 0.36 mg g$^{-1}$ Fwt, respectively, at $S_{90}$ mmol L$^{-1}$ NaCl levels in strawberry plants. The positive effects of *P. oleracea* on the Chl-*a* and Chl-*b* contents at $SP_{90}$ mmol L$^{-1}$ NaCl were evident, as the Chl-*a* and Chl-*b* contents were 1.01 and 0.51 mg g$^{-1}$ Fwt, respectively, in strawberry plants (Figure 2A,B).

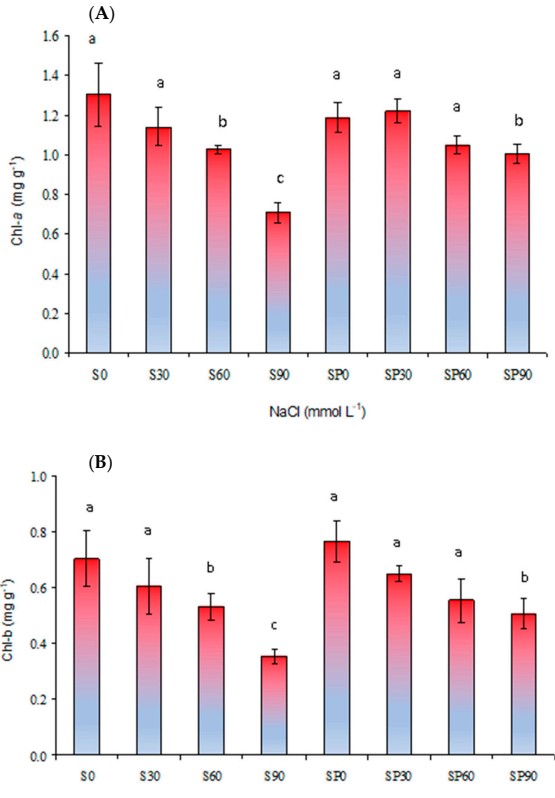

**Figure 2.** Leaf Chl-*a* (**A**) and Chl-*b* (**B**) contents of strawberry plants grown alone or in combination with *P. oleracea* at differing NaCl levels (0, 30, 60, and 90 mmol L$^{-1}$). S: a strawberry grown alone; SP: the strawberry and *P. oleracea* companionship; TSS: total soluble solids.

　　　Leaf proline content significantly increased as the concentration of NaCl levels increased as a response to salinity stress. ($p \leq 0.05$); see Figure 3A. The highest proline level was determined as 13.77 μmol g$^{-1}$ Fwt with the $S_{90}$ mmol L$^{-1}$ NaCl treatment, whereas at the $SP_{90}$ condition, the proline level decreased to 4.40 μmol g$^{-1}$. Therefore, the combination of *P. oleracea* not only improved the physiological and biochemical conditions of strawberry plants but also reduced the stress metabolite levels.

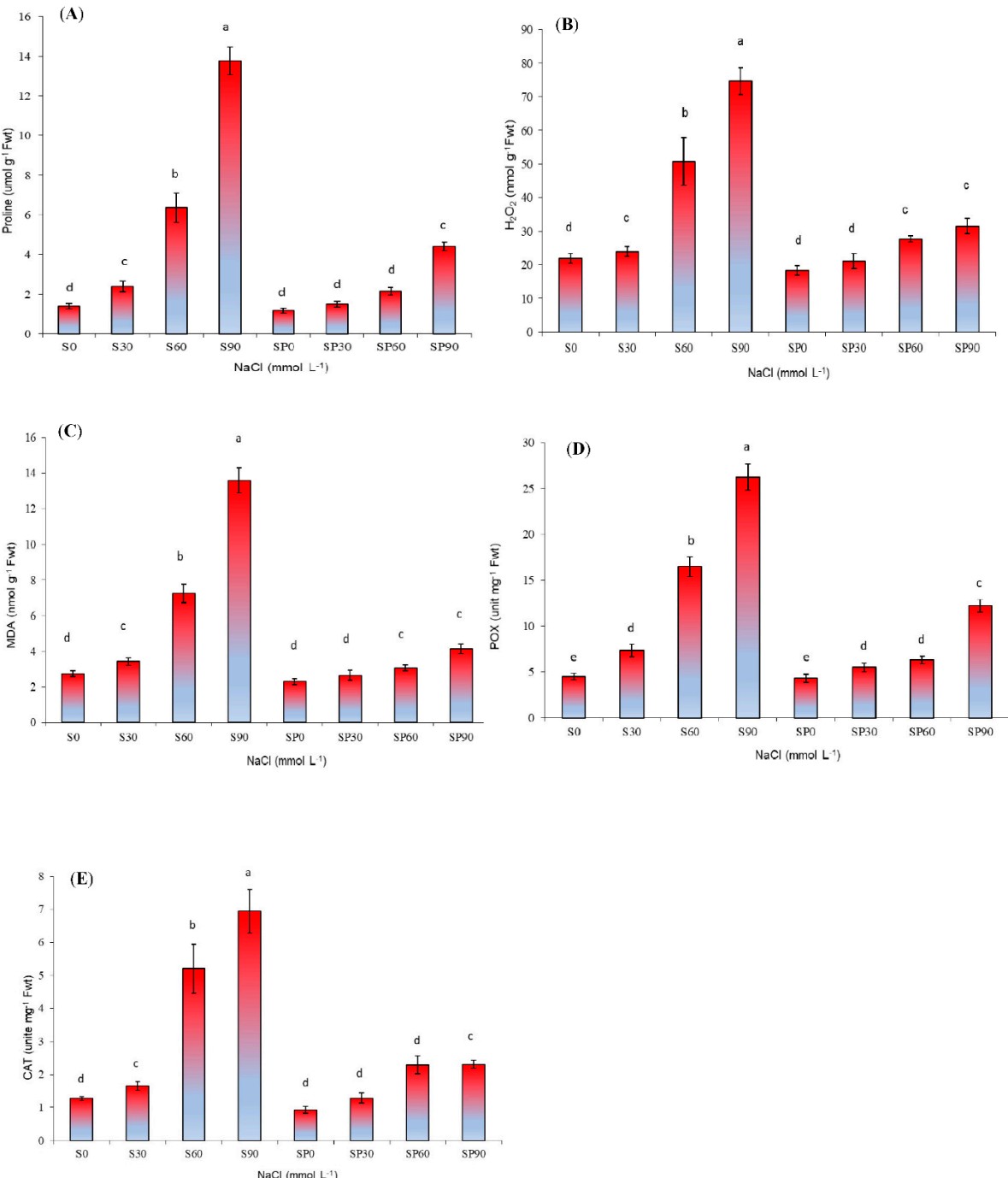

**Figure 3.** Proline (**A**), H$_2$O$_2$ (**B**), and malondialdehyde (MDA) (**C**) contents; peroxidase enzyme activity (POX) (**D**) and catalase enzyme activity (CAT) (**E**) antioxidant enzyme contents of strawberry plants grown alone or in combination with *P. oleracea* at differing NaCl levels (0, 30, 60, and 90 mmol L$^{-1}$). S: a strawberry grown alone; SP: the strawberry and *P. oleracea* companionship; TSS: total soluble solids.

The companionship of *P. oleracea* had a remarkable effect to reduce the impact of NaCl stress in strawberry plants. Again, leaf $H_2O_2$ and MDA contents increased with the increasing levels of salt stress. The highest $H_2O_2$ and MDA levels were determined as 74.72 and 13.58 nmol g$^{-1}$ Fwt, respectively, at the $S_{90}$ mmol L$^{-1}$ NaCl level. The co-cultivation of *P. oleracea* with strawberry plants reduced the contents of $H_2O_2$ and MDA levels down to 31.56 and 4.15 nmol g$^{-1}$ Fwt, respectively (Figure 3B,C).

POX and CAT antioxidant enzymes showed parallel patterns to those of previous parameters. The co-cultivation of *P. oleracea* significantly decreased antioxidant enzyme levels at the 60 and 90 mmol L$^{-1}$ NaCl conditions, (Figure 3D,E).

*Leaf Mineral Contents*

The concentrations of beneficial ions such as those of $K^+$ and $Ca^{2+}$ decreased with the increases in salinity levels in strawberry plants. The lowest $K^+$ and $Ca^{2+}$ ions were determined at the $S_{90}$ level. The leaf $Mg^{2+}$ content was not significantly affected upon NaCl stress. *P. oleracea* co-cultivation with strawberry plants enhanced the $Mg^{2+}$ ion level at NaCl treatments; see Table 3. Under saline conditions, gradual increases of $Na^+$ and $Cl^-$ ions were evident in strawberry plants grown at increasing NaCl salinity, but the employment of *P. oleracea* significantly decreased the $Na^+$ and $Cl^-$ ion contents; Table 3.

**Table 3.** Strawberry leaf mineral contents of strawberry plants grown alone or in combination with *P. oleracea* at differing NaCl levels.

| Treatments | K+ (%) | Ca2+ (%) | Mg2+ (%) | Na+ (%) | Cl− (%) |
|---|---|---|---|---|---|
| $S_0$ | 2.32 ± 0.10 [a] | 2.39 ± 0.13 [a] | 0.32 ± 0.03 [a] | 0.17 ± 0.04 [e] | 0.35 ± 0.01 [d] |
| $S_{30}$ | 1.82 ± 0.08 [b] | 1.97 ± 0.11 [a] | 0.28 ± 0.03 [a] | 0.32 ± 0.02 [d] | 0.67 ± 0.03 [d] |
| $S_{60}$ | 1.41 ± 0.06 [c] | 1.86 ± 0.04 [b] | 0.29 ± 0.02 [a] | 0.69 ± 0.05 [b] | 2.09 ± 0.29 [b] |
| $S_{90}$ | 1.02 ± 0.05 [e] | 1.72 ± 0.03 [b] | 0.28 ± 0.03 [a] | 1.09 ± 0.03 [a] | 3.69 ± 0.23 [a] |
| $SP_0$ | 2.42 ± 0.11 [a] | 2.25 ± 0.06 [a] | 0.35 ± 0.02 [a] | 0.09 ± 0.01 [e] | 0.25 ± 0.05 [d] |
| $SP_{30}$ | 2.27 ± 0.06 [a] | 2.15 ± 0.13 [a] | 0.33 ± 0.03 [a] | 0.19 ± 0.03 [e] | 0.46 ± 0.03 [d] |
| $SP_{60}$ | 1.78 ± 0.15 [b] | 2.15 ± 0.25 [a] | 0.34 ± 0.03 [a] | 0.31 ± 0.05 [d] | 1.16 ± 0.10 [c] |
| $SP_{90}$ | 1.66 ± 0.09 [b] | 2.14 ± 0.11 [a] | 0.31 ± 0.02 [a] | 0.49 ± 0.08 [c] | 1.29 ± 0.06 [c] |

Significance level at $p \leq 0.05$ was determined for the salt treatment using Duncan's multiple range test. Different letters in each column indicate statistical differences. S: a strawberry grown alone; SP: the strawberry and *P. oleracea* companionship.

## 4. Discussion

Salinity stress is one of the most devastating issues that damages crop plants in terms of quantity and quality. Increased salinity levels not only damage plants during vegetative stages but also negatively affect reproductive stages. Under salt stress, $Na^+$ is extensively accumulated in the shoots and roots of cultivars and $K^+$ content is decreased [43]. Quality parameters such as vitamin contents, aromatic substances, and pigments are remarkably reduced. Leaf proline content, as a response to stress, tends to increase. Increasing proline content under salinity conditions indicates the adverse effects of osmotic stress on the plant. Proline and soluble carbohydrates (also known as compatible solutes) are expected to be accumulated under salinity in strawberry [44]. This can be considered to be a criterion for stress tolerance [45]. This study showed that *P. oleracea* in combination with strawberry decreased proline levels under salinity along with the reduction of $Na^+$ and $Cl^-$ ion levels by reducing the toxic levels of salt ions. *P. oleracea* gave promising results on strawberry plants grown at different NaCl stress levels (0, 30, 60, and 90 mmol L$^{-1}$). It is important to note that peat has a high bulk density. For example, Nugraha et al. [46] stated that capillary water movement had a very critical role in supplying water to the rooting zones of crop plants or the top parts of the soil. They reported that the rate of capillary water movement progressively corresponded to the increase in bulk density. Farina et al. [47] also stated that NaCl accumulation in peat mulching was much lesser than that of soil. They stated that if the porosity in the surface layers became small enough, irrigation or raindrops could plug macropores in the surface. They either block main avenues for water and roots to move

through the soil or they form a cement-like surface layer when the soil dries. The rock-solid upper layer or salt crust then restricts water movement and plant emergence. In our study, evaporation in the greenhouse was not high enough to build up NaCl accumulation in the top part of the soil. Therefore, no salt crust formation, which would have affected the results of our experimental findings, was observed.

Mozafari et al. [48] stated that salinity negatively affected the growth parameters, pigment content, and membrane stability, as well as disturbing the ionic balance in plants. For example, Saied et al. [49] stated that strawberry was considered to be a saline-sensitive plant. Many physiological and biochemical parameters deteriorated. This both directly and indirectly led to diminished productivity in plants [50]. We determined that fresh weight, dry weight, stomatal conductance, fruit average weight, fruit total yield, chlorophyll (Chl-*a* and Chl-*b*), total soluble solids, lycopene content, vitamin C content, and leaf mineral content ($K^+$ and $Ca^{2+}$) of strawberry plants significantly decreased with increasing NaCl levels. Strawberry plants grown in companionship with *P. oleracea* improved the condition of plants, and much lesser reductions in terms of total yield and quality were evident. The positive effect on strawberry growth was quite remarkable. The leaf electrolyte, proline, malondialdehyde, $H_2O_2$ contents, catalase enzyme activities, nd peroxidase enzyme activities, and leaf mineral contents ($Na^+$ and $Cl^-$) of strawberry plants increased with an increasing level of salinity. The companion plants helped strawberry plants by reducing toxic ion levels, antioxidant enzyme levels, and stress metabolites. With the improvement of those parameters, electrolyte leakage and stomatal conductance were also improved, and this was reflected in the quality of fruits in terms of lycopene and vitamin C contents. This study proved that mixed planting with *P. oleracea* in saline conditions was an effective phytoremediation technique that might significantly increase the yield production and quality of strawberry. Similar findings were also made for *S. soda* plants by Karakas [51] who suggested that the improvement of tomato plants via companion plants under salt stress (1.3 and 6.5 dS m$^{-1}$) was achieved with the synthesis of substances used for fruit development instead of building up substances for mechanisms of stress tolerance. It is important to note that synthesizing stress metabolites and antioxidant enzymes is quite costly for plants to cope with abiotic or biotic stress factors [17]. Instead of generating crop plants that can combat stress factors, the strategy that involves removing stress factors would be much appreciated. Any genetic modifications or biochemical approaches that increase the removing capacity of toxic ions or compounds from the soil habitat would be an environmentally friendly approach and a safe strategy. For example, Grafienberg et al. [52] and Karakas et al. [51] stated that reductions in stress metabolites and the uptake of toxic ions enabled tomato plants to use more energy to build up organic components such as lycopene and proteins instead of producing substances for defense mechanisms. In this study, salinity stress resulted in a reduction in vitamin C content and lycopene contents in strawberry. Jamalian et al. [53] showed that salinity reduced the vitamin C content of strawberries, which was in line with the results of the present study. The decrease in the vitamin C content of fruits at high salinity levels can be attributed to the decrease in carbohydrate (sugar) production caused by the decrease in photosynthesis required for vitamin C biosynthesis.

Yaghubi et al. [44] reported that MDA concentration was also high in strawberry plants at salt stress conditions. They reported that reactive oxygen species (ROS) production was muck higher than the scavenging capacity of antioxidant enzymes. The dismutation of $O_2^{-2}$ into $H_2O_2$ and $O_2$ was reported to increase $H_2O_2$ concentration [54]. This was observed by a higher $H_2O_2$ content in salt-stressed strawberry plants than in control plants. Since $H_2O_2$ was accompanied by an increase in the key antioxidant enzymes such as CAT, POD, and superoxide dismutase (SOD), a reduction of $H_2O_2$ was achieved. In our methodology, we achieved the decrease of stress metabolites while suppressing the antioxidant enzymes via the use of *P. oleracea* plants. Though antioxidant enzymes such as CAT, POD, and SOD are known to substantially reduce the levels of $O_2^-$ and $H_2O_2$ in plants and play a vital role in plant defense against oxidative stress [55], the increase of these enzymes might

interfere with the chemical compounds involved in quality parameters such as lycopene and vitamin C. With the use of *P. oleracea*, we were able to reduce stress metabolites and toxic ions, reduced further damages to cell components, and increased the quality-related compounds without increasing defense-related antioxidant compounds. This saved the energy to be used for defense responses, and this saved energy could be used to increase metabolic functions and quality parameters.

## 5. Conclusions

Strawberry cultivation has become popular recently, and this has led to an increase in cultivated areas. These areas have become saline-polluted, saline-prone, and saline-prevalent. Since strawberry is a salt-sensitive plant, it is easily affected by a mild or moderate level of salinity. A very low level of NaCl could reduce crop yield and reduce the quality of fruits.

In this study, strawberry seedlings were grown alone or in combination with *P. oleracea* under differing NaCl concentrations. Strawberry seedlings under increasing NaCl salinity were negatively affected in terms of physiological, morphological, and biochemical parameters. Defending plants synthesized various stress metabolites such as proline, MDA, $H_2O_2$, and antioxidant enzymes to ease the negative effects of NaCl toxicity. However, increases of these metabolites were negatively correlated with quality-related metabolites such as vitamin C and lycopene. The cultivation of strawberry plants with *P. oleracea* plants reduced the concentrations of stress metabolites and antioxidant enzyme levels, as well as indirectly contributing to increases of vitamin C and lycopene contents.

We suggest that the employment of *P. oleracea* would remediate the conditions of strawberry parameters by accumulating $Na^+$ and $Cl^-$ ions, thus causing reductions in the synthesis of stress metabolites. The use of *P. oleracea* is a quite practical and environmentally friendly approach where salinity is prevalent. *P. oleracea* has a high potential that could be used in high saline and in other environmental stress conditions.

**Author Contributions:** Conceptualization, S.K., M.D. and I.B.; methodology, S.K., I.B. and M.D.; software, S.K. and M.D.; validation, S.K., I.B. and M.D.; formal analysis, S.K. and M.D.; investigation, S.K., I.B. and M.D.; resources, S.K., I.B. and M.D.; data curation, S.K.; writing—original draft preparation, S.K.; writing—review and editing, M.D. and I.B.; visualization, S.K., I.B. and M.D.; supervision, S.K.; project administration, S.K.; funding acquisition, S.K. All authors have read and agreed to the published version of the manuscript.

**Funding:** This research was funded by Harran University Scientific Research Project (HUBAP), grant number 17247.

**Institutional Review Board Statement:** Not applicable.

**Informed Consent Statement:** Not applicable.

**Conflicts of Interest:** The authors declare no conflict of interest.

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
