# Peer review of "The Use of Halophytic Companion Plant (Portulaca oleracea L.) on Some Growth, Fruit, and Biochemical Parameters of Strawberry Plants under Salt Stress"

_horticulturae, doi:10.3390/horticulturae7040063_

Round 1
Reviewer 1 Report
The authors, in this study, explore the ability of Halophytic Companion Plant P. oleacea, in pots coculture, to mitigate the effect of soil salinity on strawberry, a moderately tolerant plant to NaCl saline stress. They showed many ameliorative effects such as increased fresh weight, dry weight, and fruit average weight and total fruit yield of strawberry plants along with the improvement of physiological and biochemical parameters. After careful reading of the manuscript, this reviewer highlights some experimental shortcomings and provides suggestions.
From what was written in the materials and methods, the authors irrigated the plants for 5 months with different saline solutions. Even if the authors limited themselves to a single irrigation per week at field capacity, after 23 weeks, due to the accumulation effect, the salt concentration of the pots 90 mM NaCl should have reached 2M, that is 4 times higher than seawater i.e. incompatible concentration with the microflora of the soil and with the vital activities of the root.
The statistical analysis of the experimental data is totally missing
The introduction appears incomplete and summary in particular on the effects of osmotic stress generated by the salinity of the soil
Line 61 in 8-L pots? 0.8 L is more likely.
The authors show the data obtained through the same graphic typology that is repeated 15 times in terms of graphics and chromatism, making it boring and not very diversified for the reader.
Author Response
Thank you very much for the suggestions and criticisms about our manuscript. We performed our best to fulfil the requirements raised by the referees.
Referee No #1
Question 1 From what was written in the materials and methods, the authors irrigated the plants for 5 months with different saline solutions. Even if the authors limited themselves to a single irrigation per week at field capacity, after 23 weeks, due to the accumulation effect, the salt concentration of the pots 90 mM NaCl should have reached 2M, that is 4 times higher than seawater i.e. incompatible concentration with the microflora of the soil and with the vital activities of the root.
Answer 1. We would like to correct the misunderstanding. After 1 week of the establishment of strawberry seedlings in pots of torf, we had sawn P. oleracea seeds to the pots and we pinched the excess P. oleracea seedlings and arranged 25 seedlings per pot. After the start of establishment of the root of P. oleracea, we started irrigation with or without NaCl. This establishment of P. oleracea took almost 5 weeks. The experimental trial from the very begining of obtaining strawberry seedlings till the end of the harvest took 5 months. The actual NaCl treatment during this period lasted for 12 weeks.
Since the pots contained torf and irrigated with or without NaCl nutrient solution on demand, which is the pot capacity in our case, and drain was available under the pots, no excess solution containing NaCl, therefore, was accumulated whatsoever in any concentrations of NaCl treatment (30-, 60- and 90 mmol L-1 NaCl).
If NaCl was able to be accumulated, this would not only be visible at 90 mmol L-1 NaCl treatment, which was suggested that 2 M NaCl would be expected at the end of the trial according to the Referee No#1, but also other NaCl treatments such as 30- and 60 mmol L-1 NaCl would be accumulated at the same ratio or similar pattern, only may be much less build up of NaCl such as 1.5 or 1 M NaCl would be expected to accumulate according to the Referee No#1. If this was the case, it would have been no significant differences observed in terms of biochemical and physiological responses of strawberry seedlings, which have been regarded as moderate salt-sensitive or salt-sensitive plants in many publications. Therefore, we disagree with Referee No#1, at this point. At very high concentrations of NaCl, it would have been impossible to get any physiological and biochemical responses from strawberry seedlings. We have different responses proved by the graphs and photos. We could discuss this further and prove it firmly if further questions and raised by the Referee.
Question 2. The statistical analysis of the experimental data is totally missing
Answer 2. In the second point, the criticism was made to the statistical analysis and we added statistical analysis to the graphs and we showed the significant differences accordingly.
Question 3. The introduction appears incomplete and summary in particular on the effects of osmotic stress generated by the salinity of the soil
Answer 3. The introduction and summary were revised.
Question 4. Line 61 in 8-L pots? 0.8 L is more likely.
Answer 4. The pot volume of 8-L. There is no mistake here.
Question 5The authors show the data obtained through the same graphic typology that is repeated 15 times in terms of graphics and chromatism, making it boring and not very diversified for the reader.
Answer 5 Graphs were reorganized, and legends were improved according to Referee No#1.
Reviewer 2 Report
Karakas et. al. reported useful findings in the manuscript entitled" The Use of Halophytic Companion Plant (Portulaca oleracea L.) on Some Growth, Fruit, and Biochemical Parameters of Strawberry Plants Under Salt Stress". However, I have few comments about the manuscripts.
1) Introduction part needs attention. It is short and incomplete. Please write about the salinity problem worldwide and its effect on crop production. The authors did not explain the halophytic plant P. oleracea L. Its distribution, uses, and importance. The number of citations is also very less in the article and especially in the introduction.
2) Figure 1 SP9 should be SP90.
3) Inline, 213 correct the % sign.
4) All the figures need significant differences to be shown in letters such as a,b,c...etc.
5) Figure legends should be descriptive.
6) In the manuscript, the authors reported a decrease in antioxidants and osmolites in the salt-treated strawberries cultivated with halophytes as compared to alone-grown strawberry plants. Explanation of P. oleracea and its role in salt tolerance in the introduction part will help understand these statements in the discussion part.
Author Response
Thank you very much for the suggestions and criticisms about our manuscript. We performed our best to fulfil the requirements raised by the referees.
Question 1. Introduction part needs attention. It is short and incomplete. Please write about the salinity problem worldwide and its effect on crop production. The authors did not explain the halophytic plant P. oleracea L. Its distribution, uses, and importance. The number of citations is also very less in the article and especially in the introduction.
Answer 1. Introduction part was reorganized and completed.
Question 2. Fig 1 SP9 sould be SP90 this was corrected
Answer 2. Figures 1 was corrected SP90.
Question 3. Inline, 213 correct the % sign.
Answer 3. Necessary corrections were made as indicated.
Question 4. All the figures need significant differences to be shown in letters such as a,b,c...etc.
Answer 4. Letters in fig a, b, c, d were inserted.
Question 5. Figure legends should be descriptive.
Answer 5. Fig legends were developed.
Round 2
Reviewer 1 Report
Although the authors significantly revised the manuscript making it fluid and understandable, they do not fully answer the reviewer's questions. The question of salt in the soil still remains unclear. The problem is not finding an agreement between the authors and the reviewer, but between the authors and the numbers. It is true what the authors say, namely that in all treatments a high concentration of salt would be reached after 12 weeks of irrigation. Can the authors indicate the millimoles of salt added in the pots for each treatment over the 12 weeks? I also remember that the salt added with irrigation tends to accumulate in the first layers of soil in a high way due to the evaporation of the water before passing into the atmosphere. Therefore the distribution of the salt along the profile of the pot is not homogeneous. Did the authors take this into account? If, on the other hand, the authors are convinced that there is no strong accumulation of salt in the soil, they would provide at least a hypothesis of its fate.
Author Response
Answers to Questions Raised by the Reviewer 1
Thank you very much for the suggestions and criticisms about our manuscript. We performed our best to fulfil the requirements raised by the referees once more.
Question 1 Although the authors significantly revised the manuscript making it fluid and understandable, they do not fully answer the reviewer's questions. The question of salt in the soil still remains unclear. The problem is not finding an agreement between the authors and the reviewer, but between the authors and the numbers. It is true what the authors say, namely that in all treatments a high concentration of salt would be reached after 12 weeks of irrigation. Can the authors indicate the millimoles of salt added in the pots for each treatment over the 12 weeks? I also remember that the salt added with irrigation tends to accumulate in the first layers of soil in a high way due to the evaporation of the water before passing into the atmosphere. Therefore the distribution of the salt along the profile of the pot is not homogeneous. Did the authors take this into account? If, on the other hand, the authors are convinced that there is no strong accumulation of salt in the soil, they would provide at least a hypothesis of its fate.
Answer 1.
We would like to point out the methods and results are clearly described and presented and at the same time research design is not appropriate according to the referee. We the authors corrected and answered each point raised by the referee to the best of our knowledge.
We pointed clearly that we irrigated the pots of torf with irrigation water containing different NaCl concentrations such as 30,- 60-, and 90 mmol L-1. If there is no drain in the pots whatever the concentration of NaCl irrigation is made, NaCl would build up eventually. The irrigation period would have no matter in this case, the NaCl build-up in the pots would be inevitable. But we have drains in the pots and we irrigate the pots with the right amount of water as the pot capacity in our case. How come NaCl build-up would be expected under such conditions. We are prepared to give examples of workers that used saline irrigation water. We would like to indicate two recent articles for the notice of referee and the editor. We could give more examples.
1st article title “Potassium silicate alleviates deleterious effects of salinity on two strawberry cultivars grown under soilless pot culture”
2nd article title “Salinity stress mitigation by humic acid application in strawberry (Fragaria x ananassa Duch.)”
Referee 1 indicated that salt accumulates in the first layers of soil in a highway due to the evaporation of water.
Well, this information has no relation with our findings, we use pots of torf and the temperature around pots is not high enough to ascend the salt ions to the upper levels of pots through capillary routes. Pots contain torf; capillary routes in torf are not like the ones in the soil. Also, the temperature above pots, canopy temperature in the vicinity of pots, is not high to result in excess evaporation so we have not observed any salt accumulation in the first layer of torf in the pots. Also, we would like to point out the roots of strawberry plants are below the surface. The experimental trial was performed in a semi-controlled greenhouse in the fall season, so it is not possible to expect any high evaporation and we, of course, did not notice any evaporation that results in the accumulation of salt crust on the surface of pots which will possibly be ascended due to upward transfer of NaCl ions to the upper parts of pots. Irrigation water containing NaCl will go down to the pots and will be taken by the roots of strawberry and the roots of halophyte Portulaca oleracea. Our aim is to see if we could protect the strawberry plants via the companionship of the halophyte Portulaca oleracea plants. We proved that the halophyte plants absorbed NaCl and did not compete with strawberry plants in terms of water consumption, mineral uptake and light absorption. Our main target is to improve the condition of strawberry plants under NaCl conditions. By reducing NaCl stress on strawberry, we improved the health status of strawberry and increased the contents of vitamin C and lycopene content. Metabolites that would be synthesized for stress were reduced via the companion plants. Instead, the energy which would be synthesized for stress metabolites was spent for the synthesis of quality parameters. We also discussed our findings in the discussion section and made our conclusions supporting our results. We are very surprised that Referee 1 marked this point as “the conclusions are not supported by the result”.
We would like to point out of unnecessary discussion of NaCl distribution in the soil. We do not study soil genesis and soil physical properties. So, it would be very odd to discuss this point in our manuscript. Also, with this approach, via the use of halophytes, we have the potential to reduce the toxicity of heavy metal contaminated- or pesticide-polluted soils if we could find an appropriate plant to work with the crop plants. With this mechanism, we initially try to reduce the stress factors via the companion plants rather than increasing the tolerance of cultural plants. So, with this approach, even the susceptible plants could have a chance to be cultivated in adverse conditions.

Reviewer 2 Report
1) Figure 3 legend need attention "Proline (C), H2O2 (D), MDA (E) contents and POX (A) and), CAT (B) a)" these alphabets are not matching with the figures. 2) Also, the Figure 3D POX figure needs statistics.Author Response
Answers to Questions Raised by the Reviewer 2
Thank you very much for the suggestions and criticisms about our manuscript. We performed our best to fulfil the requirements raised by the referees once more.
Question 1 Figure 3 legend need attention "Proline (C), H2O2 (D), MDA (E) contents and POX (A) and), CAT (B) a)" these alphabets are not matching with the figures.
2) Also, Figure 3D POX figure needs statistics.
Answer 1. Necessary corrections were made as indicated, line 360-line 363.
Answer 2: The statistical corrections were made, line 3349-line 353.
